# Effects of Exergaming in Patients with Cardiovascular Disease Compared to Conventional Cardiac Rehabilitation: A Systematic Review and Meta-Analysis

**DOI:** 10.3390/ijerph19063492

**Published:** 2022-03-15

**Authors:** Carles Blasco-Peris, Laura Fuertes-Kenneally, Tomas Vetrovsky, José Manuel Sarabia, Vicente Climent-Paya, Agustín Manresa-Rocamora

**Affiliations:** 1Institute for Health and Biomedical Research of Alicante (ISABIAL), 03010 Alicante, Spain; blasco_carper@gva.es (C.B.-P.); laurafuertesken@gmail.com (L.F.-K.); jsarabia@umh.es (J.M.S.); amanresa@umh.es (A.M.-R.); 2Department of Physical Education and Sport, University of Valencia, 46010 Valencia, Spain; 3Cardiology Department, Alicante General University Hospital (HGUA), 03010 Alicante, Spain; 4Faculty of Physical Education and Sport, Charles University, 16252 Prague, Czech Republic; tomas.vetrovsky@gmail.com; 5Department of Sport Sciences, Sports Research Centre, Miguel Hernández University of Elche, 03202 Elche, Spain

**Keywords:** exercise-based cardiac rehabilitation, virtual reality, videogames, coronary artery disease, exercise capacity

## Abstract

*Background*: Exercise-based cardiac rehabilitation (CR) programs are used for improving prognosis and quality of life in patients with cardiovascular disease (CVD). Nonetheless, adherence to these programs is low, and exercise-based CR programs based on virtual reality (i.e., exergaming) have been proposed as an alternative to conventional CR programs. However, whether exergaming programs are superior to conventional CR programs in patients with CVD is not known. *Objective*: This systematic review with meta-analysis was conducted to explore whether exergaming enhances exercise capacity, quality of life, mental health, motivation, and exercise adherence to a greater extent than conventional CR programs in patients with CVD. *Method*: Electronic searches were carried out in PubMed, Embase, Web of Science, and Cumulative Index to Nursing and Allied Health Literature databases up to June 2021. Meta-analyses were performed using robust variance estimation with small-sample corrections. The effect sizes were calculated as the mean differences (MD) or standardized mean differences (SMD) as appropriate. The SMD magnitude was classified as trivial (<0.20), small (0.20–0.49), medium (0.50–0.79), or large (≥0.80). Heterogeneity was interpreted based on the *I*^2^ statistics as low (25%), moderate (50%), or high (75%). *Results*: Pooled analyses showed no differences between exergaming and conventional CR programs for enhancing exercise capacity (i.e., distance covered in the six-minute walk test) (MD_+_ = 14.07 m (95% confidence interval (CI) −38.18 to 66.32 m); *p* = 0.426) and mental health (SMD_+_ = 0.17 (95% CI −0.36 to 0.70); *p* = 0.358). The results showed a small, statistically nonsignificant improvement in quality of life in favor of exergaming (SMD_+_ = 0.22 (95% CI = −0.37 to 0.81); *p* = 0.294). Moderate heterogeneity was found for exercise capacity (*I*^2^ = 53.7%), while no heterogeneity was found for quality of life (*I*^2^ = 3.3%) and mental health (*I*^2^ = 0.0%). *Conclusions*: Exergaming seems not to be superior to conventional CR programs for improving exercise capacity, quality of life, or mental health in patients with CVD.

## 1. Introduction

Cardiovascular disease (CVD) is the leading cause of death globally [1] and its prevalence is expected to increase in future years [2]. Therefore, patients with CVD deserve special care to maintain or restore their quality of life and improve their functional capacity and psychological wellbeing. In this regard, cardiac rehabilitation (CR) is a multifaceted and multidisciplinary intervention recommended in patients with coronary artery disease (CAD) or chronic heart failure (CHF) [3]. CR programs integrate core elements such as exercise, nutrition, psychological wellbeing, education, and tobacco cessation, thanks to the work of an interdisciplinary team that includes physicians, psychologists, dietitians, nurses, and physical and rehabilitation medicine (PRM) specialists [4]. Moreover, it is cost-effective as it reduces hospitalization and health care expenditures while improving prognosis [5]. Accordingly, it is recommended in international guidelines with the highest level of scientific evidence—class IA—for all patients with CAD or CHF [6,7].

Within CR programs, exercise-based CR is the intervention, backed by scientific evidence, that best contributes to a decrease in morbidity, mortality, and hospital readmissions. In addition, there is evidence showing that exercise-based CR programs improve quality of life and exercise capacity [8,9,10], which has been proven to be an independent predictor of mortality and cardiovascular events in patients with CHF and CAD [11,12]. The six-minute walk test (6MWT) is one of the most widely used parameters to assess exercise capacity in exercise-based CR programs [13]. A systematic review and meta-analysis found that patients that participated in a CR program improved, on average, 60.43 m in the 6MWT [14]. In addition, the 6MWT has been demonstrated in previous studies to correlate well with peak oxygen uptake (VO_2_ peak), which is considered the “gold standard” for assessing exercise capacity [11,12].

Despite the growing evidence, based on the benefits of CR, its implementation and adherence are suboptimal and vary amongst rehabilitation centers in Europe [15]. CR referral rates in different studies varied from 22 to 74%. For referred patients with CVD, participation rates ranged from 14 to 35% with dropout rates of 12 to 56% [16]. CR adherence ranged from 36.7 to 84.6% of sessions with a mean adherence of 66.5 ± 18.2% (median 72.5%). It must be noted that CR adherence is significantly lower among females than males [17]. In recent years, there has been a growing interest in the use of new technologies as a possible solution to this problem. One of these promising tools is virtual reality (VR), defined by the Encyclopedia Britannica (Lowood) as the use of computer modeling and simulation that enables a person to interact with an artificial three-dimensional (3D) visual or another sensory environment. VR-based exercises, also known as exergames, are fun and engaging, helping to improve adherence and overcome some traditional exercise barriers such as lack of motivation [18]. Their benefits for improving motor function recovery, manual dexterity, and fall prevention have been demonstrated in patients with stroke, Parkinson’s disease, and older adults [19,20,21,22]. The high cost of these technologies has limited their implementation in everyday practice in the past, but now more affordable options have emerged, such as the XBOX^®^ One console or the Nintendo Wii^®^ console.

Previous studies have been performed to examine the beneficial role of CR programs based on exergaming. For instance, Verheijden Klompstra, Jaarsma, and Strömberg [23], who carried out a scoping review in older adults, concluded that exergaming could increase physical activity in patients suffering from cardiac disease and is safe and feasible in older adults. These authors also found positive outcomes in relation to balance, cognitive performance, and depressive symptoms. Therefore, exergaming seems to be suitable for this population. Nonetheless, whether exergaming is superior to conventional CR programs (i.e., exercise-based CR or usual care) is not clear yet. Previous reviews reported that VR supplementation significantly increased physical activity when compared with conventional CR [24]. Conversely, no significant differences were observed between interventions in energy expenditure, quality of life, psychosocial parameters, or clinical outcomes [25]. Davis, Parker, and Gallagher [26] showed similar results regarding physical activity, with an improvement of motivation in patients who underwent the VR exercise program compared to usual care. In the same line, previous reviews, which included studies performed with stroke patients, also failed to find differences between exergaming and conventional CR programs for improving daily living activities [19,22]. Finally, the largest systematic review carried out to date with patients with CVD did not provide a meta-analysis of the results and included patients with cardiovascular risk factors who had not yet developed CVD [27]. Moreover, further evidence has come to light since the publication of García-Bravo’s review [28,29], and so an update of the subject is needed.

In order to address this need, a systematic review with meta-analysis was conducted (a) to summarize the previous evidence regarding exercise-based CR programs combining physical activity with VR, videogames, and/or other technology (i.e., exergaming), and (b) to determine whether exergaming enhances exercise capacity, quality of life, mental health, motivation, and exercise adherence to a greater extent than conventional CR programs (i.e., exercise-based CR or usual care) in patients with CVD. Based on previous evidence, we hypothesize that, even though exergaming could be suitable in patients with CVD, there will be no differences between exergaming and conventional CR for improving exercise capacity, quality of life, and mental health. Nonetheless, VR adds a motivational factor through entertainment that could be an advantage over conventional CR programs. Therefore, we hypothesize that exergaming will enhance patients’ motivation and, therefore, their adherence to the exercise regime.

## 2. Methods

The present review was performed following the Preferred Reporting Items for Systematic reviews and Meta-analyses (PRISMA) guidelines [30]. The protocol was prospectively registered in the PROSPERO database (CRD42021285596).

### 2.1. Data Search and Sources

Electronic database searches were performed in PubMed, Embase, Web of Science, and Cumulative Index to Nursing and Allied Health Literature databases (CINAHL) up to June 2021. Free-text terms related to patients and interventions were used to carry out electronic searches. The full search strategy can be found in the Appendix A. The reference lists of previous systematic reviews and meta-analyses, as well as of the included studies, were hand-reviewed to identify potential studies which fulfilled our inclusion criteria. Moreover, authors of the included studies were emailed to identify unpublished studies to diminish the impact of the publication bias in our findings.

### 2.2. Study Selection

Eligibility criteria were established according to the PICOS (participants, interventions, comparisons, outcomes, and study design) guideline as follows: (a) Participants: studies which included adult patients with CVD (e.g., patients with CHF, CAD, or who have undergone revascularization (i.e., coronary artery bypass grafting or percutaneous coronary intervention)), regardless of the sex. Studies that included patients with other pathologies or congenital CVD were excluded; (b) Intervention: studies where patients allocated to the experimental group (EG) performed an exercise-based CR program which combined physical activity/exercises with videogames, VR, or other technologies (i.e., exergaming). Studies in which the EG only performed relaxation and/or respiratory exercises were excluded; (c) Comparisons: uncontrolled and controlled studies which included an active (i.e., exercise-based CR) and/or passive (i.e., usual care) control group (CG), where technology was not used; (d) Outcomes: exercise capacity, quality of life, mental health, motivation, and exercise adherence were selected as endpoints of interest; and (e) Study design: randomized and nonrandomized studies were included. Only studies written in Spanish or English were included. Where two or more articles referred to the same study, only one article was included in the review.

Two authors (C.B. and L.F.) assessed all identified titles/abstracts for possible inclusion and reviewed the full texts against the inclusion criteria. In case of disagreement, a third author (A.M.) assessed the study to reach an agreement.

### 2.3. Data Extraction and Coding Study Characteristics

Two authors (C.B. and L.F.) coded the characteristics of the included studies using a standardized data extraction form. Disagreements were solved by a third author (A.M.).

The following information was extracted from the included studies: (a) study characteristics (journal, publication year, study design (i.e., uncontrolled study, randomized controlled study, or nonrandomized controlled study), and country); (b) participant characteristics (sample size, sex (i.e., males, females, or mixed sample), men percentage, age, CVD diagnosis (i.e., HF, CAD, or mixed sample), and risk factors/comorbidities (e.g., hypertension, diabetes, and dyslipidemia)); (c) intervention and comparison characteristics (setting (i.e., supervised, home-based, or mixed), phase of exercise-based CR, type of exercise, intervention length, training frequency, intensity, type of comparison group (i.e., active or passive), and technology information); and (d) outcome information. Authors were contacted via e-mail to obtain lacking information.

### 2.4. Risk of Bias Assessment

The RoB 2 tool (revised tool for risk of bias in randomized studies) and the ROBINS-I tool (risk of bias in nonrandomized studies of interventions) were used for assessing the risk of bias in randomized and nonrandomized controlled studies, respectively [31].

### 2.5. Computation of Effect Size and Statistical Analyses

Separate meta-analyses were performed for the domains of exercise capacity, quality of life, mental health, motivation, or exercise adherence when reported in at least three studies. For the domains including only one measure, we expressed the effect size as the mean difference (MD). For the domains including more measures, we used the standardized mean difference (SMD) calculated using the Hedges’ g statistic. The magnitude of SMD was classified as trivial (<0.20), small (0.20–0.49), medium (0.50–0.79), or large (≥0.80) [32]. In studies with more than one CG, the sample size of the EG was split up by the number of CGs to avoid overinflation of the sample size [33]. Robust variance estimation, a form of random-effects models, was used to carry out pooled analyses, allowing us to include multiple effect sizes from the same study [34,35]. Heterogeneity was assessed using the *I*^2^ index [36] and classified as low, moderate, or high at 25%, 50%, and 75%, respectively [37]. All analyses were performed using packages robumeta (version 2.0) and metafor (version 2.0-0) in R version 3.4.4 (The R Foundation for Statistical Computing, Vienna, Austria).

## 3. Results

### 3.1. Study Selection

Figure 1 illustrates the systematic review process. In brief, from a total of 1524 studies after deleting duplicates, 20 studies were eligible for full-text analysis after reviewing titles and abstracts. After reviewing the full texts, eight studies were included [24,28,29,38,39,40,41,42], and 12 were excluded for the following reasons: no exercise-based CR intervention (*n* = 9) [22,43,44,45,46,47,48,49,50], patients with other pathologies were also included (*n* = 1) [51], the patients included were the same as another publication and study design did not fulfill our inclusion criteria (*n* = 1) [52], and lack of information/abstract (*n* = 1) [53]. Although efforts were made to localize unpublished studies, all included studies had been published in peer-reviewed journals.

### 3.2. Study Characteristics

Study and participant characteristics are summarized in Table 1. The eight included studies are from 11 countries and were published from 2006 to 2021. Seven studies (87.5%) were randomized controlled studies [24,28,29,38,39,41,42] and one (12.5%) was an uncontrolled study with pre-post design [40]. The studies enrolled 733 patients with CVD (384 allocated to the EGs and 349 to the CGs). Four studies (50.0%) recruited patients with CAD [38,39,41,42], two studies (25.0%) recruited patients with CHF [29,40], and two studies (25.0%) recruited both patients with CAD and CHF [24,28]. The sample size in the EGs ranged from 10 to 234 patients, with a mean ± standard deviation age of 58.5 ± 6.9 years (min–max: 48.7–66.0 years), while the sample size in the CGs varied from 10 to 230 patients, with a mean ± standard deviation age of 59.5 ± 4.9 years (min–max: 52.0–67.0 years). The total sample size ranged from 20 to 464 patients. Two studies (25.0%) recruited exclusively male patients [39,41] and six (75.0%) included male and female patients, but most of them were males [24,28,29,38,40,42].

Intervention characteristics are summarized in Table 2. Out of seven controlled studies, five (71.4%) included an active comparator group [24,28,38,39,42], one (14.3%) a passive control group [29], and one (14.3%) included two comparator groups (active and passive) [41]. Five studies (62.5%) carried out a supervised exercise-based CR program [24,28,38,39,42], two (25.0%) a home-based program [29,40], and one (12.5%) carried out a combined exercise-based CR program (supervised and home-based training sessions) [41]. One study (12.5%) performed an inpatient exercise-based CR program (phase I) [38] and seven (87.5%) carried out an outpatient exercise-based CR program, of which, four and three studies performed a phase II [24,28,39,42] and III [29,40,41] exercise-based CR program, respectively. The intervention duration, which was reported in six studies, ranged from six to 48 weeks. Four studies (50.0%) performed less than four training sessions a week [24,39,41,42], two (25.0%) performed more than three sessions a week [29,40], one (12.5%) carried out two sessions a day [38], and one (12.5%) did not report this information [28]. Regarding technology used to carry out training sessions in the EGs, four studies (50.0%) used oculus glasses or other 3D technology [28,38,39,41], three (37.5%) used a game console [24,29,40], and one (12.5%) used several devices [42]. The remaining information about training sessions (e.g., session length and intensity) can be found in Table 2.

### 3.3. Risk of Bias Assessment

All controlled studies were randomized, and the assessment of the risk of bias, which is shown in Figure 2, was performed by using the RoB 2 tool. The overall risk of bias was judged as high in all included studies. Biases arising from the randomization process and the intended intervention were the most frequent domains causing downgrading. The risk of bias of the uncontrolled study was considered as high due to lack of comparability.

### 3.4. Outcome Measures

Four of the controlled studies measured exercise capacity using the distance covered in the 6MWT. Three controlled studies, one of them including two comparator groups, reported quality of life (measured by the MacNew questionnaire or the SF-36 questionnaire) and mental health (measured by the Beck-II Depression questionnaire or the Hospital Anxiety and Depression scale). Therefore, pooled analyses were carried out for exercise capacity, quality of life, and mental health, while the remaining findings reported in the included studies, which did not reach the minimum number of studies required to perform meta-analyses, can be found in Table 2.

### 3.5. Pooled Analyses

Meta-analysis showed statistically nonsignificant difference between exergaming and conventional CR programs in exercise capacity changes (measured as the distance covered in the 6MWT) (number of analysis units (k) = 4; MD_+_ = 14.07 m (95% CI = −38.18 to 66.32 m); *p* = 0.426; Figure 3). Heterogeneity was moderate for exercise capacity (*I*^2^ = 53.7%). On the other hand, our findings showed a small, statistically nonsignificant improvement in quality of life in favor of exergaming compared to conventional CR programs (k = 11; SMD_+_ = 0.22 (95% CI = −0.37 to 0.81); *p* = 0.294; Figure 4). Finally, no difference was found between exergaming and conventional CR for enhancing mental health (k = 5; SMD_+_ = 0.17 (95% CI = −0.36 to 0.70); *p* = 0.358; Figure 5). No inconsistency was found for quality of life (*I*^2^ = 3.3%) and mental health (*I*^2^ = 0%).

## 4. Discussion

This systematic review with meta-analysis was conducted to summarize the previous evidence regarding exercise-based CR supplemented with VR (i.e., exergaming) and determine whether it enhances exercise capacity, quality of life, mental health, motivation, and exercise adherence to a greater extent than conventional CR programs (i.e., with or without exercise) in patients with CVD (i.e., CAD or CHF). In accordance with our hypothesis, the pooled analyses showed that exergaming does not improve these variables in patients with CVD compared to conventional CR programs. These findings are similar to those previously reported in patients with CVD and other pathologies. For instance, Fang, Wu, Lv, Chen, and Zeng [19] and Zhang, Li, Liu, Wang, and Xiao [22] found no differences in exercise capacity between exergaming and conventional CR programs in stroke patients. Similarly, Radhakrishnan, Baranowski, Julien, Thomaz, and Kim [25] concluded that, even though exergaming seems to be appealing for older adults, it did not affect the quality of life or mental health (i.e., anxiety and depression) of patients with CVD. Therefore, our findings seem to support the theory that exergaming is not superior to conventional CR programs for enhancing exercise capacity, quality of life, or mental health in patients with CVD.

Regarding exercise capacity, although a trend was found in favor of exergaming for improving the distance covered in the 6MWT, the results were not statistically significant nor clinically relevant. The MD in 6MWT between the exergaming and the conventional CR group obtained in this analysis (14.07 m) was lower than the minimal clinically important difference of 25 to 27 m reported in subjects with CAD [54] and the 35 to 37 m in patients with CHF [55]. In addition, Garcia-Bravo, Cano-de-la-Cuerda, Dominguez-Paniagua, Campuzano-Ruiz, and Barrenada-Copete [42] and Ruivo, Karim, O’Shea, Oliveira, and Keary [24] did not find any differences in the metabolic equivalent of task between the exergaming and conventional CR programs. These findings are in line with the previously reported reviews in stroke patients [19,22]. The only study in our systematic review that found differences in exercise capacity between the exergaming and the conventional CR exercise-based program was Cacau, Oliveira, Maynard, Araújo Filho, and Silva [38], who performed a phase I CR program (during hospitalization). Interestingly, the timing of the intervention in relation to the index event and the stage of the disease might account for these positive results. On this subject, Klompstra, Jaarsma, and Strömberg [40] found that patients with a lower New York Heart Association class and shorter time since diagnosis were more likely to improve their exercise capacity in the 6MWT. This data confirms the assumption that the earlier the better when considering the initiation of CR, which has been widely reported [56,57,58,59,60]. Not only is CR more effective when delivered in earlier stages of the disease, but it is also more effective when started promptly after the cardiac event. Any delays are directly related to worse outcomes [61,62]. Nonetheless, it should be noted that Cacau, Oliveira, Maynard, Araújo Filho, and Silva [38] did not assess the patients’ baseline exercise capacity, and only compared both groups postintervention, thus, there could be intergroup differences regarding baseline exercise capacity that may be interfering with the results. Jaarsma, Klompstra, Ben Gal, Ben Avraham, and Boyne [29] exemplified the importance of detecting these differences in patients’ baseline characteristics to correctly interpret the data. Initially, Jaarsma, Klompstra, Ben Gal, Ben Avraham, and Boyne [29] also found a higher exercise capacity in favor of exergaming at three, six, and 12 months compared to conventional CR without exercise; however, this effect was lost when they corrected for preintervention 6MWT values (patients in the exergaming group had a higher baseline 6MWT than the control group). Many factors could influence a patient’s 6MWT. Not surprisingly, Klompstra, Jaarsma, and Strömberg [40] pointed out that the number of comorbidities a patient presented significantly impacted their 6MWT at baseline and should also be considered. Therefore, whether exergaming is better than conventional CR programs for enhancing exercise capacity remains uncertain, and future research is needed to determine whether the phase in which exergaming is initiated (early versus late) has an impact on the outcomes.

There are many reasons that could account for the inconsistent results regarding exercise capacity we have mentioned above. The studies included in our review displayed moderate heterogeneity for exercise capacity, but we could not analyze the influence of potential moderator variables on the effects of training due to the low number of studies. These moderator variables include the intensity of the workout, the length of the intervention, or the type of VR technology used in each program, which could all explain the heterogeneity of our findings. Firstly, there is evidence showing that the intensity of a workout is one of the most important determinants of the physiological response to training [63]. Nonetheless, most of the studies included do not control the intensity of the exercise-based CR programs nor the associated energy expenditure of the workouts. Only three of the studies supervised the intensity during training sessions [24,39,41]. According to Peng, Lin, and Crouse [64], exergaming produces physiological and hemodynamical effects similar to, or even greater than, conventional CR, but the intensity of the exercise should be at least moderate to achieve results. Nonetheless, evidence shows that the intensity of exergames can vary widely from 2.0 to 4.2 metabolic equivalent of task [64] and could account for the inconsistent results. Therefore, it is important to monitor and control the intensity of the training sessions using heart rate or the rate of perceived exertion. Secondly, Peng, Lin, and Crouse [64] disclosed that the type of VR technology used could also impact the effect training has on exercise capacity. In this regard, the studies included in our systematic review showed no consensus on the type of exergaming they used. Half of the studies used complex 3D devices such as oculus glasses [28,38,39,41] that are expensive and not easily available, while the other half employed game consoles [24,29,40] which are much more affordable and accessible. Consequently, the influence of the type of VR technology used on the training-induced effect should be addressed in future studies. In addition, the risk of bias assessment showed methodological limitations. Most of the studies included in the review did not report the method they used to create the randomization sequence. Moreover, due to the nature of the studies, neither the patients, personnel who delivered the intervention, nor the assessors were blinded to the patient’s allocation. If the patients and/or personnel are aware of their group allocation, it is more likely that additional health-related behaviors will differ between the groups. This could induce deviation from the intended intervention and affect the results. Our systematic review also revealed that there was a great heterogeneity regarding the number of training sessions per week (i.e., training frequency), the intervention duration, or the phase of CR in which the intervention was carried out. All the issues previously disclosed could explain, at least in part, the heterogeneity observed in our findings regarding the effect of exergaming on exercise capacity compared with conventional CR programs in patients with CVD.

According to the American Heart Association Exercise/American Association of Cardiovascular and Pulmonary Rehabilitation, one of the objectives of CR programs is to achieve emotional wellbeing [65]. In this regard, our findings showed no differences between exergaming and conventional CR programs for enhancing the quality of life or mental health; even though a trend was found in favor of exergaming for improving quality of life, the magnitude of the effect was small. These results concur with those previously reported by Radhakrishnan, Baranowski, Julien, Thomaz, and Kim [25], although controversial findings have been reported in patients with stroke. For instance, Domínguez-Téllez, Moral-Muñoz, Salazar, Casado-Fernández, and Lucena-Antón [66] found a statistical increase in quality of life in favor of exergaming. It should be noted however, that these authors included studies that used other tools (i.e., modified Barthel index and functional independence measure) to measure patients’ wellbeing. There is evidence that shows that the number of comorbidities could also be affecting the quality of life and mental health scores of these patients [67]. To a certain degree, this could account for the inconsistent results compared to stroke patients. Conventional CR programs without exergaming, however, have already been proven to increase the quality of life of their participants [8]. Therefore, it is challenging to detect statistically significant differences between conventional CR and exergaming, as in most studies we revised, both groups usually perform the same exercise regime with the only difference being the use of VR. In summary, a small improvement in quality of life was found in our pooled analysis in favor of exergaming, but future studies should be performed to determine whether this improvement is greater than the one achieved with regular CR programs. Finally, despite the high heterogeneity found in our systematic review regarding the intervention and patient characteristics, quality of life and mental health showed no heterogeneity.

One aspect that could be contributing to the improvement in quality of life seen in exergaming is social interaction. Various studies have highlighted the role of VR in stimulating social interaction among its participants, with the benefits being maintained beyond the completion of the program. Ruivo, Karim, O’Shea, Oliveira, and Keary [24] reported that 80% of patients considered themselves to be more talkative with their peers and had a higher social score in the McNew quality of life questionnaire. Klompstra, Jaarsma, and Strömberg [40] mentioned that exergaming supplementation in elderly subjects could facilitate a greater connectedness with family, especially grandchildren. Moreover, the challenges that involved having a teammate and/or competing with other teams were more motivating than those without [26], and social support through the involvement of spouses or grandchildren facilitated adherence [25]. By harnessing the social support VR provides, we could potentially increase motivation and adherence in participants of CR programs. Higher exercise adherence is key for improving the prognosis of patients with CVD. Previous studies have shown that patients who completed a CR program had a 31% reduction in cardiac mortality [68], obtaining a 1% reduction in mortality per session [69]. In light of the prognostic benefits of CR, the European Society of Cardiology urges the creation of a model that optimizes the result of long-term CR programs.

Based on this evidence, we hypothesized that exergaming could enhance patients’ motivation and, as a consequence, their adherence to exercise. However, our expectations were not met in most of the studies included in our review. Only Ruivo, Karim, O’Shea, Oliveira, and Keary [24] reported a lower discontinuation rate in the exergaming group compared to the CG (6.3% vs. 18.8%, respectively). In contrast, Gulick, Graves, Ames, and Krishnamani [28] and Jaarsma, Klompstra, Ben Gal, Ben Avraham, and Boyne [29] reported a higher dropout rate in those patients allocated to the exergaming group compared to the CG. Regarding motivation, Jaarsma, Klompstra, Ben Gal, Ben Avraham, and Boyne [29] measured it directly through a validated motivation questionnaire and showed no differences between the two groups for improving motivation. In the same line, the study of da Cruz, Ricci-Vitor, Borges, da Silva, and Turri-Silva [51], which was excluded from our systematic review due to the inclusion of patients with cardiovascular risk factors, found that adding one session of exergaming to a conventional CR program (i.e., two weeks) significantly diminished motivation in patients with CVD or cardiovascular risk factors. Conversely, Garcia-Bravo, Cano-de-la-Cuerda, Dominguez-Paniagua, Campuzano-Ruiz, and Barrenada-Copete [42], who used a nonvalidated client satisfaction questionnaire, reported that patients found exergaming to be more motivating than conventional CR programs. As we can see, few studies have analyzed the effect of exergaming on motivation and exercise adherence, and their findings are controversial. Due to the low number of studies and heterogeneity of the results, we could not include in the meta-analysis the results of motivation and exercise adherence. Therefore, future studies should be performed to properly design CR programs based on VR for enhancing motivation and exercise adherence in patients with CVD.

Some issues regarding patient characteristics, which could limit the scope of our findings, should be addressed. The mean age of the patients in the exergaming and conventional CR groups was 58.5 and 59.5 years, respectively. It is well known that CHF mainly affects older people, with incidence and prevalence rising steeply with age: from around 1% for those aged <55 years to >10% in those aged 70 years or over [70,71]. In addition, Puymirat, Simon, Cayla, Cottin, and Elbaz [72] reported that the mean age of patients with ST-segment-elevation myocardial infarction and non-ST-segment-elevation myocardial infarction was 63 and 68 years, respectively. Therefore, our study population is younger than the patients seen in clinical practice and our results cannot be extrapolated to older patients. Nevertheless, previous studies, such as Agmon, Perry, Phelan, Demiris, and Nguyen [73], have already proven that exergaming is safe and feasible in this age group.

Most of the patients selected in the studies included in our systematic review were males, highlighting the underrepresentation of females in scientific literature and CR programs. Despite studies suggesting that females have a higher risk of morbidity and mortality after an acute coronary event [74] and that they achieve similar benefits from CR to males, females are paradoxically less likely to be referred to CR (39% vs. 45%) [75] and, once enrolled, have lower adherence rates. Of the included studies, Klompstra, Jaarsma, and Strömberg [40] found a difference between males and females regarding the time spent exergaming (i.e., males generally favor exergaming more than females). Thus, further efforts should be made to increase the representation and adherence of females in these programs.

## 5. Strengths and Limitations

This is the first systematic review that has been performed exclusively with patients with CVD. Moreover, pooled analyses were performed to analyze the effect of exergaming compared with conventional CR programs. Nonetheless, there are some limitations that should be disclosed. First, the low number of studies included did not allow us to carry out a meta-analysis of some of the included outcomes (i.e., motivation and exercise adherence). Secondly, all of the studies included were previously published, and therefore could increase the risk of publication bias. Finally, heterogeneity, publication bias, and sensitivity analyses were not performed due to the low number of included studies.

## 6. Conclusions

Our findings have demonstrated that exergaming does not enhance exercise capacity, quality of life, mental health, motivation, or exercise adherence to a greater extent than conventional CR programs. These findings should be considered with caution due to the low number of included studies. Moreover, our conclusions should be limited to young male patients with CVD. Future studies should include more elderly patients and females with CVD to determine whether the phase of CR or training variables, such as the intensity, duration, and type of technology used in the workouts, impacts the effects of exergaming.

## Figures and Tables

**Figure 1 ijerph-19-03492-f001:**
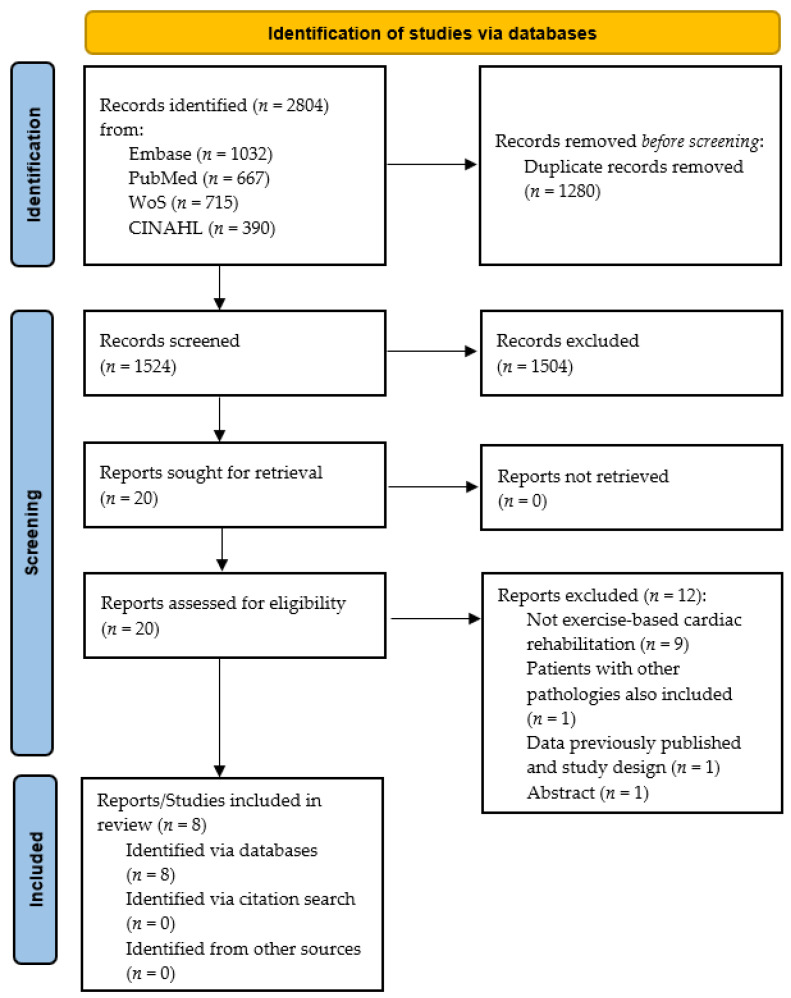
Flow chart of the systematic review process.

**Figure 2 ijerph-19-03492-f002:**
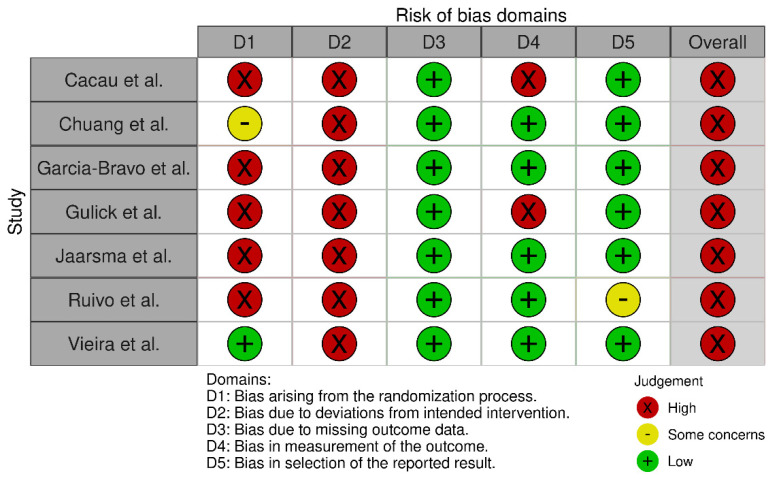
Risk of bias of the controlled studies rated using RoB 2 tool.

**Figure 3 ijerph-19-03492-f003:**
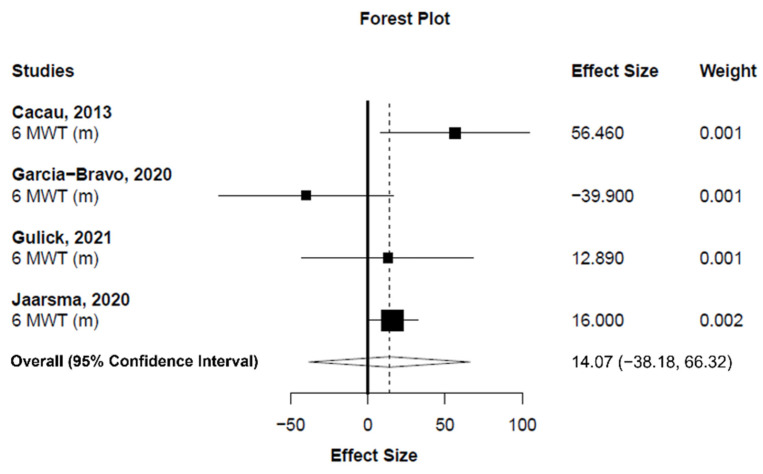
Forest plot showing the mean difference between exergaming and conventional cardiac rehabilitation for six-minute walk text. Mean difference higher than zero favors exergaming.

**Figure 4 ijerph-19-03492-f004:**
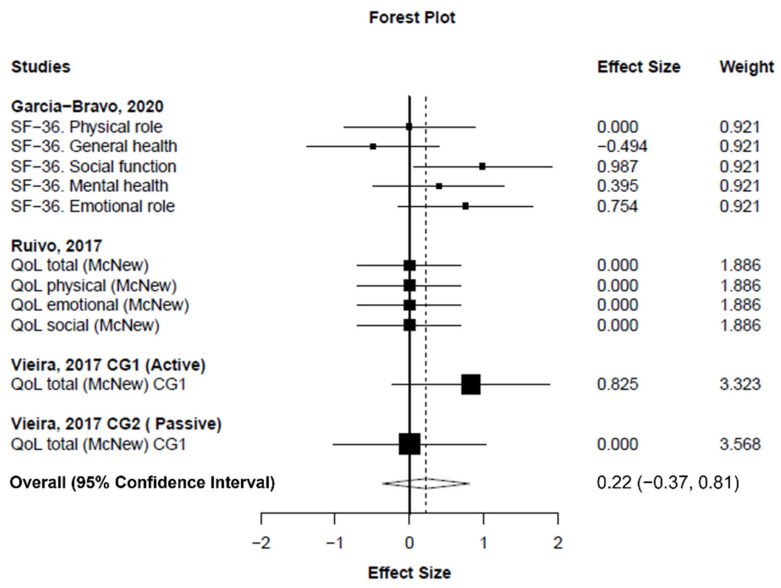
Forest plot showing the standardized mean difference between exergaming and conventional cardiac rehabilitation for quality of life. Standardized mean difference greater than zero favors exergaming. CG: control group; SF-36: short-form 36 health survey questionnaire.

**Figure 5 ijerph-19-03492-f005:**
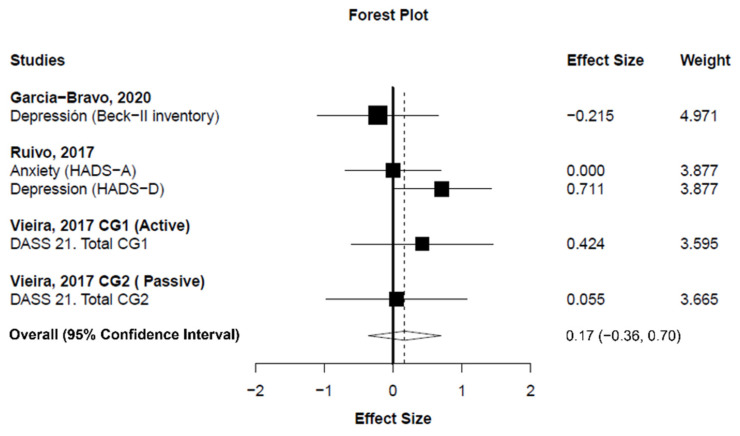
Forest plot showing the standardized mean difference between exergaming and conventional cardiac rehabilitation for mental health. Standardized mean difference greater than zero favors exergaming. CG: control group; HADS: Hospital Anxiety and Depression Scale.

**Table 1 ijerph-19-03492-t001:** Study and participant characteristics.

Study (Author, Year)	Group	Study Characteristics	Participant Characteristics	
		Country; Study Design; Journal	Sample Size; Male Percentage; Age	CVD Diagnosis; Risk Factors or Comorbidities
Cacau et al. [38] (2013)	EG	Brazil; RCT; Rev Bras Cir Cardiovasc	30; 43.0%; 49.2 ± 2.6 years	CAD; NR
	CG		30; 53.0%; 52.0 ± 2.4 years	
Chuang et al. [39] (2006)	EG	Taiwan; RCT; Phys Ther	10; 100%; 65.7 ± 14.5 years	CAD; AHT, DM2, DLP, SM
	CG		10; 100%; 63.7 ± 10.3 years	
Garcia-Bravo et al. [42] (2020)	EG	Spain; RCT; Int J Environ Res Public Health	10; 70.0%; 48.7 ± 6.7 years	CAD; NR
	CG		10; 100%; 53.7 ± 10.3 years	
Gulick et al. [28] (2021)	EG	USA; RCT; J Med Internet Res	41; 72.0% *; 61 ± 9.9 years *	MS; COPD
	CG		31; 72.0% *; 61 ± 9.9 years *	
Jaarsma et al. [29] (2021)	EG	Sweden, Italy, Israel, Netherlands, Germany, USA; RCT; Eur J Heart Fail	234; 72.0% ^; 66 ± 12.0 years	CHF; AHT, AF, CVA, COPD, DM2, MI
	CG		230; 70.0% ^; 67 ± 11.0 years	
Klompstra et al. [40] (2014)	EG	Sweden; Single Intervention; BMC Geriatrics	32; 68.8%; 63.0 ± 14.0 years	CHF; SM
Ruivo et al. [24] (2017)	EG	Ireland; RCT; J Cardiopulm Rehabil Prev	16; 87.5%; 59.4 ± 11.8 years	MS; AHT, DLP; DM2, OB, SM
	CG		16; 75.0%; 60.4 ± 8.5 years	
Vieira et al. [41] (2018)	EG	Portugal; RCT; Disabil Rehabil: Assist Technol	11; 100%; 55.0 ± 9.0 years	CAD; AHT, DLP; DM2, OB, SM
	CG1		11; 100%; 59.0 ± 11.3 years	
	CG2		11; 100%; 59.0 ± 5.8 years	

AHT, arterial hypertension; AF, atrial fibrillation; CAD, coronary artery disease; CG, control group; CHF, chronic heart failure; COPD, chronic obstructive pulmonary disease; CVA, cerebrovascular accident; DLP, dyslipidemia; DM2, diabetes mellitus 2; EG, experimental group; MI, myocardial infarction; MS, mixed sample; NR, no reported; RCT, randomized controlled trial; SM, smoking; OB, obesity; *, data not disaggregated by groups; ^, data reported in the pretreatment measure. Values are reported as mean ± standard deviation unless otherwise is stated.

**Table 2 ijerph-19-03492-t002:** Intervention characteristics and main findings.

Study	Intervention Characteristics	Intervention and Technology Description	Main Findings
Cacau et al. [38]	Supervised training; phase I; intervention length (NR); 2 sessions a day until hospital discharge; intensity (NR)	EG: Physiotherapeutic protocols: breathing exercises, airways clearance techniques, metabolic exercise, and motor exercise using VR CG (active): Physiotherapeutic protocols: breathing exercises, airways clearance techniques, metabolic exercise, and motor exercise	The EG had lower hospitalization length (EG: 9.5 ± 0.5 days; CG: 12.2 ± 0.9 days), as well as higher exercise capacity (6MWT) at post-intervention The EG had higher functional independence, better energy levels, and less pain, while no between-group differences were found in emotional reactions, physical ability, and social interaction (measured with the Nottingham Health Profile)
Chuang et al. [39]	Supervised training; phase II; 12 weeks; 2 sessions a week; <30 min or >30 min depending on the subject’s condition; 3 min of low intensity with progressive increase until they reach a score of 16 in Borg scale or target HR or V0_2_	EG: Treadmill with speed alteration and incline adjustments using Microsoft Direct 3D-constructed “virtual runner” model with “wraparound” screens CG (active): Treadmill with speed alteration and incline adjustments	The number of sessions needed to reach the target of 85% heart rate max and the target of 75% VO_2_ peak was lower in the EG than in the CG. Moreover, the maximum work rate achieved in the endurance training sessions was higher in the EG
Garcia-Bravo et al. [42]	Supervised training; phase II; 8 weeks; 2 sessions a week; 60 min a session; intensity adapted according to the limits of HR and sensation of effort	EG: warm-up (10 min), VR-based training (20 min), resistance exercise (endless belt) for 10 min and limb strength exercises with weight of 0.5–3.0 kg (10 min) and cool-down (10 min) CG (active): warm-up (10 min), aerobic exercise (treadmill for 30 min) and limb strength 0.5–3.0 kg (10 min) and cool-down (10 min)	No between-group differences in exercise capacity (metabolic equivalent of task and 6MWT), functional independence measure, recovery of heart rate after 6MWT, quality of life (Short Form Health Survey-36 Questionnaire), depression (Beck-II Depression Inventory), and satisfaction (Client Satisfaction Questionnaire). Moreover, no differences were found in adherence and adverse events during the intervention
Gulick et al. [28]	Supervised training; phase II; intervention length (NR); training frequency (NR); intensity (NR)	EG: Standard of care CR: 4 types of exercise equipment, including bionautica trail system (VR), stationary bikes, ellipticals, and hand rowing machines CG (active): Standard of care CR: 4 types of exercise equipment, including treadmills, stationary bikes, ellipticals, and hand rowing machines	Patient attendance was lower in the EG (58%) than in the CG (81%), with no correlation between the group and reasons for ending No between-group differences in education (5-question test), satisfaction (6-question examination), engagement (3-question test), and exercise capacity (6MWT)
Jaarsma et al. [29]	Home based; phase III; 48 weeks; 5 sessions a week; 30 min per session; intensity (NR)	EG: Standard practice at their referring center (usual care) and Nintendo Wii Sports with baseball, bowling, boxing, golf, and tennis CG (passive): Standard practice at their referring center (usual care): protocol-based physical activity advice from a heart failure team member	No between-group differences in exercise capacity (6MWT) at 3, 6, and 12 months, as well as in exercise motivation (15-question exercise motivation index), exercise self-efficacy (6-question exercise self-efficacy questionnaire), and self-reported physical activity (single item question)
Klompstra et al. [40]	Home based; phase III; 12 weeks; 7 sessions a week; 20 min per session; intensity (NR)	EG: Nintendo Wii sports. Advice 20 min everyday: bowling, tennis, baseball, golf, and boxing games	Exercise capacity (6MWT) increased from 501 ± 95 m to 521 ± 101 m. Fifty-three percent of the patients increased the distance more than 30 m, which was considered clinically relevant Lower New York Heart Association scale and shorter time since diagnosis (less than one year) were related to the increase in exercise capacity
Ruivo et al. [24]	Supervised training; phase II; 6 weeks; 2 sessions a week; 60 min per session; intensity monitored with individual target HR zones	EG: Aerobic, resistance, and flexibility training using 9 circuit stations Nintendo Wii sports (boxing and canoeing) CG (active): Aerobic, resistance, and flexibility training using 9 circuit stations and music video	Lower tendency for dropping out in the EG (6%) than in the CG (19%). Higher improvement in energy expenditure in the EG compared to the CG No between-group differences in the median individual attendance and the number of patients experiencing adverse events during the intervention. Moreover, no differences in changes in exercise capacity (metabolic equivalent of task), affect toward exercise (Positive and Negative Affect Scale), anxiety and depression (Hospital Anxiety and Depression Scale), and quality of life (MacNew)
Vieira et al. [41]	Mixed; phase III; 24 weeks; 3 sessions a week; 60 min per session approx.; two progressive levels of intensity: level 1 (65% of HR reserve) and after three months, level 2 (70% of HR reserve); intensity monitored with the Borg scale	EG: Education on cardiovascular risk factors and 10 exercises: a warm-up exercise, 7 exercises of conditioning workout aimed at enhancing muscular endurance and/or strength, and 2 exercises to increase limb flexibility using a computer and Kinect-rehab play CG1 (active): Education on cardiovascular risk factors and 10 exercises: a warm-up exercise, 7 exercises of conditioning workout aimed at enhancing muscular endurance and/or strength, and 2 exercises to increase limb flexibility using a paper booklet CG2 (passive): Usual care: Education on cardiovascular risk factors and daily walks encouraged	The EG showed an enhanced selective attention and conflict resolution ability (Stroop Test) in comparison with the two CGs In contrast, no between-group differences were found in the quality of life (MacNew), depression, anxiety, and stress (Depression, Anxiety, and Stress Scale 21)

6MWT, six-minute walk test; CG, control group; EG, experimental group; HR, heart rate; NR, no reported; V0_2_, oxygen uptake; VR, virtual reality.

## Data Availability

The datasets generated from the current review are available from the corresponding author on reasonable request.

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
