# Peer review of "Effects of Exergaming in Patients with Cardiovascular Disease Compared to Conventional Cardiac Rehabilitation: A Systematic Review and Meta-Analysis"

_ijerph, 2022, doi:10.3390/ijerph19063492_

Round 1
Reviewer 1 Report
I read with great interest this systematic review and meta-analysis entitled "Effects of exercise in patients with cardiovascular disease 2 compared to conventional cardiac rehabilitation." The article is of scientific interest and is in line with the aims of the journal. The author's guidelines have been respected, and the manuscript does not require a revision of the English language.
Introduction
The introduction argues well for the topics covered. However, there are some issues that need the authors' attention.
Page 1, lines 42-44: A reference regarding cardiac rehabilitation is missing. Please add
Page 2, lines 54-56: A reference regarding the six-minute walk test is needed. Please add
The authors avoided stating the key role of physical and rehabilitation medicine (PRM) physicians in several cardiac rehabilitation interventions. Please add
The aim of the study was made clear that exergaming could be suitable in patients with CVD. There will be no differences between exergaming and conventional CR for improving exercise capacity, quality of life, and mental health.
References should be increased. Please also consider:
-Advanced role and field of competence of the physical and rehabilitation medicine (PRM) specialist in Contemporary Cardiac Rehabilitation. Hellenic J Cardiol. 2016; 57:16-22. https://pubmed.ncbi.nlm.nih.gov/26856196/
Materials and methods
The materials and methods section adequately describes the authors' work.
Results and discussion
The results are written in a very fluid and readable way for the reader.
The discussion correctly identifies the results.
Conclusions
The conclusions are compared to the results obtained.
References
The references are relevant to the topic and recent. Add the suggested references.
Reviewer 2 Report
See the attached manuscript with my comments and items I would recommend being addressed. My main thought to the authors would be, if you hypothesize no improvements (line 108-109), why do the review? I do not see a strong argument for why this study was done with the exception of a few new articles came out.

Author Response
Author’s reply to the reviewer 2.
We thank the reviewers and the assistant editor for the helpful advice and recommendations regarding the manuscript entitled “Effects of exergaming in patients with cardiovascular disease compared to conventional cardiac rehabilitation: a systematic review and meta-analysis”, with ID ijerph-1613586.
Comments and suggestions for authors
See the attached manuscript with my comments and items I would recommend being addressed. My main thought to the authors would be, if you hypothesize no improvements (line 108-109), why do the review? I do not see a strong argument for why this study was done with the exception of a few new articles came out.
Author response
We want to thank the reviewer for the helpful advice and recommendations regarding this manuscript. Regarding your questions and suggestions, you can find the answers and corrections that have been done highlight in another file (please see attachments).
In regards to your question, even though we hypothesized that there will be no differences between exergaming and conventional CR, we still found interesting to review the subject because there was reasonable doubt whether exergaming could be superior based on the results of previous studies (mentioned in the manuscript). Some of these studies had shown improvements in exercise capacity, quality of life, motivation and/or adherence with exergaming where as others did not. Due to the inconsistent results found in the literature, we wanted to conduct a review on the subject. Moreover, this is the first systematic review with meta-analysis that has compared the effects of exergaming and conventional CR.
Although our review showed no differences between both interventions, we still consider that the results are interesting and relevant. It demonstrates that exergaming is safe and feasible in patients with CVD and, at least, equal to conventional CR programs regarding the potential benefits. Thus, exergaming could be used as an alternative to conventional cardiac rehabilitation programs. In addition, in this review we did not explore the long-term adherence to physical exercise with exergaming, which could differ from conventional CR and should be evaluated in future studies.

Reviewer 3 Report
This study is a very important topic that has been researched by the authors in an appropriate manner and I believe that the results are valuable.
II think the discussion is appropriate and contains many important implications.
However, I think there is one part that should be changed for publication.
In "Conclusions" in "Abstruct", there is a phrase "even though a trend was found in favor of exergaming for enhancing the quality of life." (line 36), but I think it is not statistically significant and therefore not desirable.
Similarly, in "6. Conclusion," there is an expression "even though a trend was found in favor of exergaming for improving quality of life" (lines 467-468), which I think may also mislead readers.
If the results are not statistically significant, I believe that it is more appropriate to state the results as they are.
Please consider making these corrections.
Reviewer 4 Report
This study sought to determine whether cardiac rehabilitation (exergaming) using a virtual reality (VR) system is superior to conventional cardiac rehabilitation in terms of exercise tolerance (6-minute walk test), quality of life (SF-36 or McNew test), and mental health (depression scale). Eight studies were pooled and no statistically significant differences were found in any of the items under consideration. From these results, the authors conclude that the Exergaming has no advantages over the conventional methods for benefits obtained from cardiac rehabilitation. As the authors state, limitation includes the small number of articles incorporated in this meta-analysis, the bias in the study participants, and the heterogeneity of the training programs used. While not sufficient for a fair evaluation of the results, it may be worth noting that the comparison was made with the conventional cardiac rehabilitation group and not with the non-exercise group. There is accumulating evidence that conventional cardiac rehabilitation not only improves exercise tolerance and quality of life in patients with heart disease, but also improves disease prognosis, such as re-exacerbation of heart failure and cardiac death. Based on the findings observed in this study, it seems reasonable to expect that rehabilitation using the VR system may have the same prognostic effects as conventional cardiac rehabilitation.
Although the authors' discussion of the results is generally conservative, my personal impression is that the results of this study could be considered more positive in the context of opening up a new option for cardiac rehabilitation.
Except for the large volume of the discussion, there are few areas where I have concerns about the manuscript as a whole, and I do not think that major revisions are necessary. I would like to point out only a few points to note.
Line 63; correct unnecessary double space. (Lines 107, 308, 349, 417, 421 as well)
Line 65: Please indicate the references shown in (15).
Line 96: These two references are both for stroke patients, is it appropriate to indicate them as CR programs?
Line 162: Please standardize to American English to match the rest of the text (and line 427 as well).
Line 362: Blinding of patients and CR providers does not seem feasible for this type of intervention trial and should be included in the study limitations. The meaning of "deviation from the intended intervention" is not clear and needs more explanation.
Line 371: "Exercise" is unnecessary.
Figures 2 and 4: "Ruivo" is misspelled in the cited authors.
Figures 3 to 5: It would be easier to understand if the results of the pooled analysis were written in the effect size column with 95% confidence intervals.
Round 2
Reviewer 2 Report
I am satisfied with the resubmission and alterations. Great work - I look forward to seeing the manuscript published.